METHODS

# GenoTriplo: A SNP genotype calling method for triploids

**Julien Roche[1,2], Mathieu Besson[2], François Allal[3], Pierrick Haffray[2], Pierre Patrice[2], Marc Vandeputte[3], Florence Phocas[1]***

**1** Université Paris-Saclay, INRAE, AgroParisTech, GABI, Jouy-en-Josas, France, **2** SYSAAF (French Poultry and Aquaculture Breeders Technical Centre), Rennes, France, **3** MARBEC, University of Montpellier, CNRS, Ifremer, IRD, INRAE, Palavas-les-Flots, France

* florence.phocas@inrae.fr

**Data Availability Statement:** The GenoTriplo package is available on the CRAN at https://cran.r-project.org/web/packages/GenoTriplo/index.html The data supporting the article are available at: https://doi.org/10.57745/7IMQDS.

## Abstract

Triploidy is very useful in both aquaculture and some cultivated plants as the induced sterility helps to enhance growth and product quality, as well as acting as a barrier against the contamination of wild populations by escapees. To use genetic information from triploids for academic or breeding purposes, an efficient and robust method to genotype triploids is needed. We developed such a method for genotype calling from SNP arrays, and we implemented it in the R package named GenoTriplo. Our method requires no prior information on cluster positions and remains unaffected by shifted luminescence signals. The method relies on starting the clustering algorithm with an initial higher number of groups than expected from the ploidy level of the samples, followed by merging groups that are too close to each other to be considered as distinct genotypes. Accurate classification of SNPs is achieved through multiple thresholds of quality controls. We compared the performance of GenoTriplo with that of fitPoly, the only published method for triploid SNP genotyping with a free software access. This was assessed by comparing the genotypes generated by both methods for a dataset of 1232 triploid rainbow trout genotyped for 38,033 SNPs. The two methods were consistent for 89% of the genotypes, but for 26% of the SNPs, they exhibited a discrepancy in the number of different genotypes identified. For these SNPs, GenoTriplo had >95% concordance with fitPoly when fitPoly genotyped better. On the contrary, when GenoTriplo genotyped better, fitPoly had less than 50% concordance with GenoTriplo. GenoTriplo was more robust with less genotyping errors. It is also efficient at identifying low-frequency genotypes in the sample set. Finally, we assessed parentage assignment based on GenoTriplo genotyping and observed significant differences in mismatch rates between the best and second-best couples, indicating high confidence in the results. GenoTriplo could also be used to genotype diploids as well as individuals with higher ploidy level by adjusting a few input parameters.

**Funding:** The work was partly supported financially by the European Maritime and Fisheries Fund (EMFF) and the French Government (FranceAgrimer) in the R&D project HypoTemp (action n° P FEA470019FA1000016) to MB, PH, PP, and FP. The funders had no role in study design, data collection and analysis, decision to publish, or preparation of the manuscript.

**Competing interests:** The authors have declared that no competing interests exist.

## Author summary

To cultivate plants, fish and shellfish more profitable for both farmers and consumers, one can utilize individuals with three chromosome sets instead of the two found in fertile populations that are diploids. These individuals, called triploids, are generally sterile and then often exhibit higher growth and quality of products, such as seedless fruits or better flesh quality for fish and shellfish. To be able to improve performances of the sterile triploids by selective breeding, it is important to know the versions of the genes present in the three chromosome sets of triploids. Until now, few methods existed to identify these three versions, and none have been demonstrated as sufficiently effective. It is the reason why we developed the GenoTriplo software. We demonstrate in this paper the possibility to accurately genotype triploids, as well as how it can be used to reconstruct pedigree information of triploid progeny. Ultimately, we expect that it can help select for reproduction the parents that have the best triploid progeny for the traits of interest such as growth, vigour or product quality.

## Introduction

Polyploidy, characterized by the presence of three or more sets of chromosomes in the nucleus, is a phenomenon that occurs spontaneously across various taxa in the tree of life, spanning from plants [1–3] to vertebrates [4]. Certain forms of polyploidy, such as triploidy, exhibit noteworthy attributes relevant to agricultural practices. Triploid individuals, possessing three sets of chromosomes, are generally sterile, impeding the production of sexual tissues and yielding favourable outcomes for farmers. In horticulture, the cultivation of seedless fruits is facilitated by the sterility of triploids, a characteristic appreciated by consumers [5]. Triploidy has also been reported to enhance growth rate and vigour in plants [6]. In aquaculture, triploid fish demonstrate an accelerated growth rate due to the energy savings stemming from the lack of sexual maturation [7]. Additionally, the enhanced flesh quality of triploid fish and shellfish is attributed to the prevention of gonadal maturation [8,9]. From an environmental perspective, the sterility of triploids serves as a barrier against the contamination of wild genotypes by selectively bred genotypes in instances of contact between these populations [10]. Triploidy also can act as a safeguard against theft of genetic progress among competing producers.

The induction of triploidy has been achieved in various plant species [11], like citrus [5] and mulberry [12], as well as in shellfish such as oysters [13] and in finfish, in particular rainbow trout [14,15].

While triploids present advantages over diploids, their widespread production in aquaculture necessitates that selective breeding programs consider their specific performance. Breeding programs obviously require fertile broodstock, and are thus performed with diploid selection candidates. In order to maximize genetic gains on desired traits for triploid production however, it would be necessary to incorporate the performance of triploids sibs in the evaluation of breeding values. Indeed, evaluating only diploid performance may be suboptimal as the genetic correlation for the same trait between diploids and triploids may differ from unity [16–18]. In mixed-family aquaculture breeding programs, families are mixed at hatching and their pedigree is recovered *a posteriori* using genomic markers [19]. In such designs, selecting for triploid performance implies to be able to genotype triploids and recover their pedigree, in order to be able to rank diploid selection candidates using breeding values from their triploid sibs.

Technically, two platforms, Illumina and Affymetrix, have been used for genotyping SNP arrays in both diploid [20] and polyploid species [21]. As reported by [21], genotype calling is complicated for polyploids because these species have more possible genotypes at a SNP locus than diploid species do (homozygote with reference allele, heterozygote, and homozygote with alternative allele). Theoretically, the number of genotypes can be up to p+1 in a species with a ploidy level of p (i.e. 4 in triploids, 5 in tetraploids, . . .). So far, genotype calling software accompanying genotyping platforms cannot identify more than 5 clusters for Illumina and 3 clusters for Affymetrix. More specifically, the GenomeStudio software from Illumina is able to provide 5 clusters, but it requires manual adjustment of the cluster boundaries for each marker, which is impractical to use for SNP arrays with several tens of thousands SNP. The Axiom Analysis Suite (AXAS) software, widely used in both plant and fish species, is only designed for genotype calling on diploid luminescence output files from the Thermo Fisher Affymetrix platform, and does not currently support triploids. Up to 2020, there were only two publicly available software, fitTetra and ClusterCall, initially written for tetraploids [22], which could call up over three genotypes using output files with allelic signals from SNP array genotyping platforms. Another software, SuperMASSA, was written for genotype calling from Genotype-By-Sequencing data for all ploidies [22]. Many methods struggle with low-frequency genotypes [23] or lack permissiveness when faced with allelic signal shifts in polyploids [24,25]. For autopolyploids, such as induced triploids in aquaculture, the major complication is distinguishing between different allele dosages (AAA, AAB, ABB, BBB), as in this case only two alleles per locus are normally present in their diploid parents.

Therefore, limited options for genotype calling in triploids exist [26] and open source tools are even more rare. As far as we know, only the R package fitTetra, initially developed for tetraploid individuals [27,28], has been implemented in a more advanced version of the package called fitPoly to consider any other level of auto-polyploidy. However, our first trial yielded some inconsistent results using fitPoly to genotype triploids in rainbow trout. Therefore, the first objective of this study was to devise a clustering method for a better genotype calling of triploid individuals and to compare our results to those of fitPoly genotype calling on our rainbow trout study case. The second objective was to implement and disseminate this new method through an R package deposited on the CRAN to ensure its free accessibility.

## Materials & methods

### Available dataset

To develop this novel genotype calling method for triploids, we used the allelic signals produced by Thermo Fisher Affymetrix platform for a French research project on genomic selection in rainbow trout [29]. The experimental stock was established from 190 dams and 98 sires of a commercial selected all-female line of Aquaculteurs Bretons breeding company (Plouigneau, France) and 1232 triploid offspring and the 190 dams and 98 sires were genotyped for 57,501 SNPs using the medium-density Rainbow Trout Axiom 57K SNP array from Thermo Fisher [30]. We retained the allelic signals for 38,033 high quality markers present in both SNP array [31,32]. Luminescence values of probsets A and B ($S_A$ and $S_B$) for each marker and individual were obtained through the AXAS software.

### Clustering algorithm

The clustering process aimed at grouping individuals that share the same genotype. To enhance the efficiency of the clustering method, variable(s) given to the algorithm must be chosen carefully so the different genotypes are well separated along the axe(s) [25]. In our approach, we decided to use 2 variables (and so 2 axes): the contrast (Eq 1) and the signal

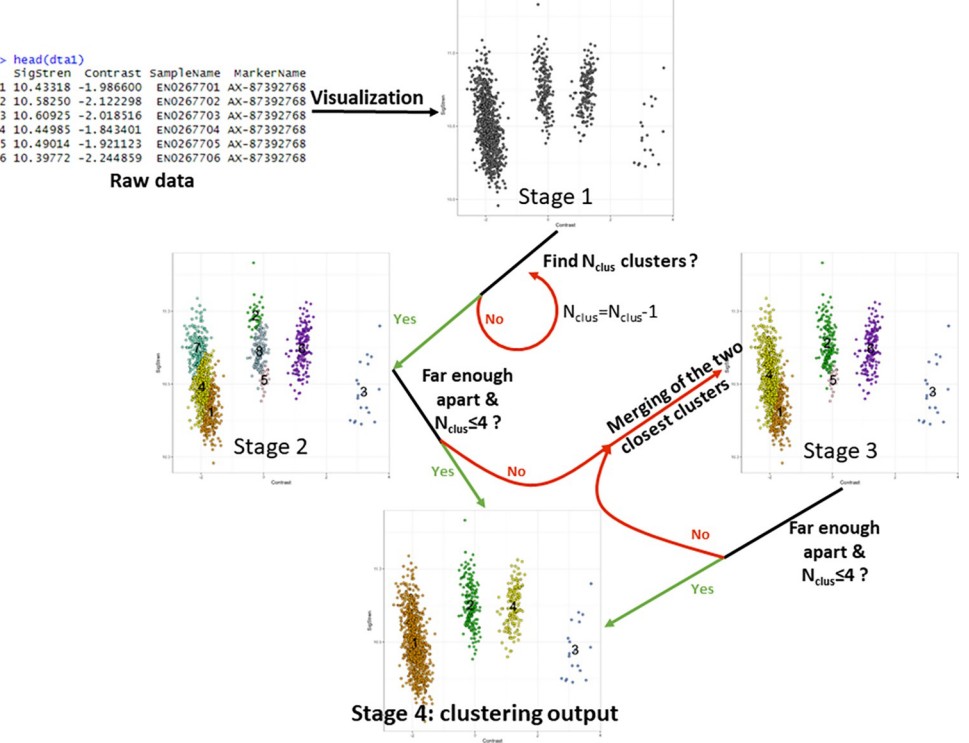

**Fig 1. Algorithm stages for the clustering phase.**

strength (Eq 2), commonly used by AXAS for diploids.

$$x = Contrast = \log_2\left(\frac{S_A}{S_B}\right) \tag{Eq1}$$

$$y = Signal\ Strength = \frac{\log_2(S_A) + \log_2(S_B)}{2} \tag{Eq2}$$

Thus, each individual was represented by a pair of coordinates (x, y) for each marker (Fig 1, Stage 1). For each SNP, the Rmixmod clustering package (version 2.1.8) [33] was then used on R software (version 4.3.1) [34] to find clusters among individuals for a given marker, with no prior information. The clustering function of Rmixmod initiates the process by randomly picking individuals as starting point and uses an expectation-maximization algorithm (EM) to probabilistically update parameters of the clusters (mean, variance, weight). $N_{init}$ initializations were performed and the one that maximized likelihood passed to the next steps.

During the initialization phase, the clustering function of Rmixmod was asked to find $N_{clus}$ clusters among individuals with $N_{clus}$ greater or equal to the number of possible genotypes for a given SNP (4 in our case) (Fig 1, Stage 2). $N_{clus}$ values of 4, 8 or 12 were tested to find an optimal value.

When the algorithm failed to find $N_{clus}$ clusters among individuals (failure of the EM algorithm to converge with $N_{clus}$ clusters), it was restarted with $N_{clus} = N_{clus}-1$ clusters and so on, until the algorithm converged and a non-error solution was obtained. For these retries, $N_{init}$ was automatically reduced by 2 (with a minimum value of 1) to limit computing time. Indeed, when the algorithm failed to find the initial number of $N_{clus}$ clusters, it was likely that the

marker did not display all possible genotypes. Thus, a high $N_{init}$ was not necessary to find a suitable solution.

As the final $N_{clus}$ might be higher than the maximum number of genotypes, a single genotype could be divided into different clusters. If more than 4 clusters remained (the maximum number of genotypes in triploids), or if two clusters were too close to be considered as distinct genotypes, the two clusters with the weakest distance in Contrast value were merged into a single one (Fig 1, Stage 3 to Stage 4). Two clusters were declared as too close if:

$$D_{Clus1,Clus2} < 0.28 * \left( 1 + abs\left( \frac{Contrast_{clus1} + Contrast_{clus2}}{2} \right) \right) \quad \text{(Eq3)}$$

$$Where, Contrast_{clusi} = Mean(x_{indiv_{clusi}}) \quad \text{(Eq4)}$$

$$And, D_{clus1,clus2} = abs(Contrast_{clus1} - Contrast_{clus2}) \quad \text{(Eq5)}$$

Where $D_{Clus1,Clus2}$ represented the distance between the centre of cluster 1 and the centre of cluster 2 in Contrast value (abscissa), and $Contrast_{Clusi}$ represented the mean Contrast value of cluster i. As the standard deviation along the Contrast axis of a genotype increased when $Contrast_{Clusi}$ moved away from 0 (to positive or negative value), the distance criteria to merge clusters had to increase the more $Contrast_{Clus1}$ and $Contrast_{Clus2}$ differed from 0.

The threshold value 0.28 in Eq 3 used to merge clusters corresponding to a same genotype was determined graphically (see Fig 2). The algorithm was forced to find 8 clusters for each marker. Then we computed, for each marker, the distance in contrast ($D_{Clus1,Clus2}$) and the absolute mean contrast ($mC_{Clus1,Clus2}$ = abs($Contrast_{clus1}$+ $Contrast_{clus2}$)/2) between each pair of clusters. For a given marker, various situations can occur:

- case a: two clusters are close to each other (low value of $D_{Clus1,Clus2}$), meaning that they likely represent the same genotype and their mean contrast value is either strongly positive (AAA genotype), positive and close to 0 (AAB genotype), negative and close to 0 (BBA genotype) or really negative (BBB genotype). We can see those four genotype groups in Fig 2 representing the distance between two clusters as a function of their contrast mean.

- case b: two clusters are distant (high value of $D_{Clus1,Clus2}$), meaning that they likely represent two different genotypes. Those clusters should not be merged and their coordinates on the y_axis of their representation in Fig 2 are higher than the coordinates of representation of clusters that should be merged (case a). Different groups of points can be detected as illustrated in Fig 2. At y-axis values ranging from 1 to 2, 3 groups detected representing from left to right, distances between BBB and BBA clusters; BBA and BAA clusters; and BAA and AAA clusters. At higher values on y-axis (from 2 to 3), 2 groups can be found representing distances between BBB and BAA clusters and between BBA and AAA clusters. Finally, at the top of the y-axis, there is the group representing the distance between BBB and AAA clusters with a mean contrast value close to 0.

As we could visually assess on the graph the clusters that should and should not be merged as representatives of different genotypes, we extracted values of distance (D) and mean contrast (M) from the separation between same genotype clusters and different genotype clusters (red line in Fig 2) and derived the coefficient as Dmin = D/(1+M) (from Eq 3). The mean of Dmin was found to be 0.28.

To assess the impact of the number of initializations i.e. random starting points on the final clustering, the algorithm was tested with three modalities for $N_{init}$: 1, 5 and 10 different initializations.

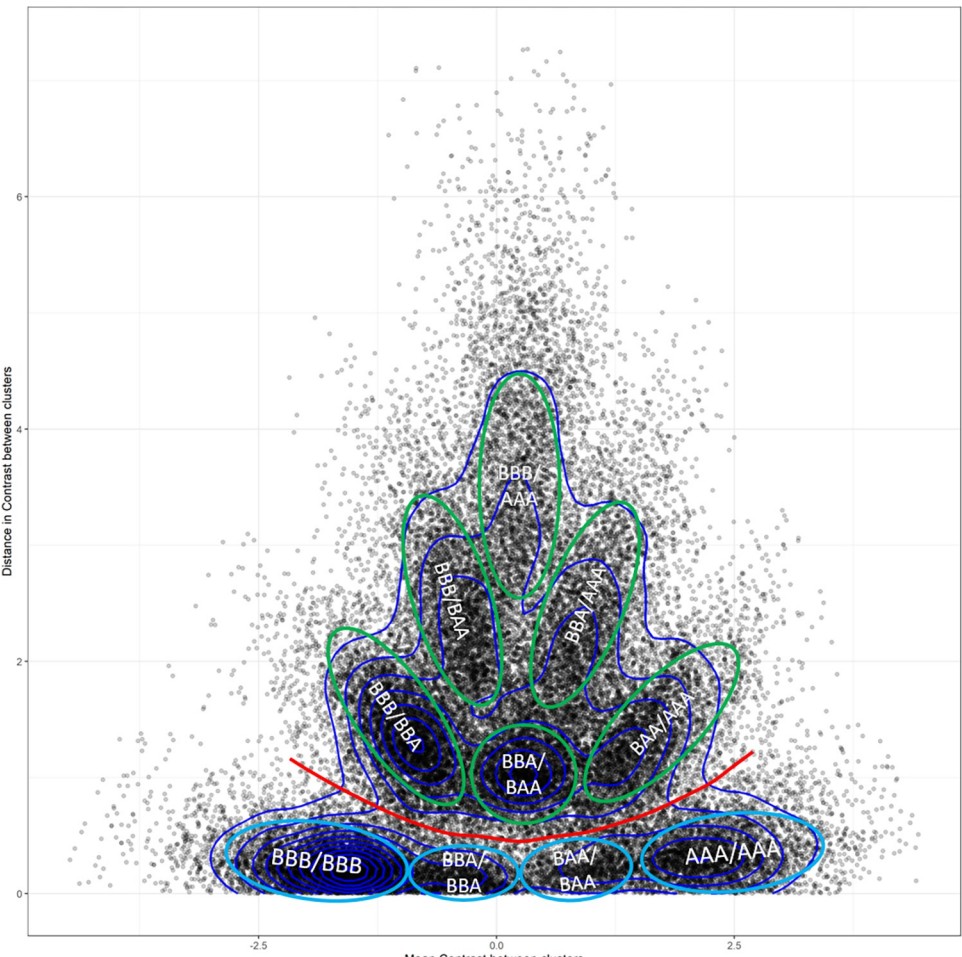

**Fig 2. Mean contrast and distances between clusters from 1803 random markers (50.000 points on the graph).**
Surrounded in blue, clusters that should be merged. Surrounded in green, clusters that should not be merged. The red line represents the separation between clusters that should and should not be merged. BBB/BBA stands for: mean contrast and distance between a supposed BBB cluster and BBA cluster.

The algorithm was also tested for three other modalities to assess the impact of $N_{clus}$ on the outcome: 4, 8 and 12, i.e. a number greater or equal to the number of possible genotypes for a given SNP (4 in our case). Other existing methods for genotyping usually look for a maximum number of clusters which exactly corresponds to the number of possible genotypes. However, by increasing the initial number of clusters (8 and 12), we aimed to enable the algorithm to identify clusters gathering only a few individuals, which can happen frequently in case of a low frequency genotype.

## Genotype calling

Two situations must be accounted for to assign genotypes to clusters depending on the origin of the samples: i) either all samples originated from a same population or ii) they come from various populations that can be genetically distant. The right situation must be specified to our algorithm as they involve different hypotheses. In our case, the samples originated from a single population, and we only used the corresponding method for genotype calling.

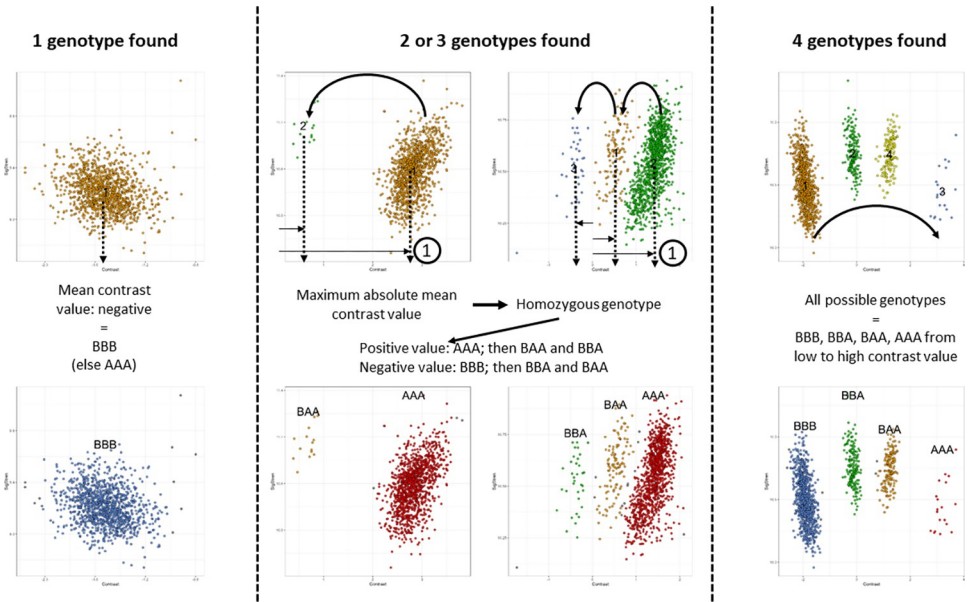

**Fig 3. Illustration of genotype determination for 1; 2 or 3; and 4 clusters identified for a given SNP.**

In the situation of a unique population, genotypes were attributed by considering the mean Contrast of each cluster and its position relative to other clusters. The most extreme cluster, identified by the absolute value of its contrast mean (x), was designated as a homozygous genotype (AAA if mean(x)>0 and BBB if mean(x)<0) (Fig 3). Other clusters were ordered by their mean contrast values, and genotypes were subsequently assigned based on the first cluster that had been assigned (Fig 3). For example, if the mean contrast was positive for the most extreme cluster (i.e. assigned as AAA), genotypes were then assigned depending on their mean contrast values in the order AAB, ABB and BBB, from the closest to the furthest cluster from the AAA homozygous genotype. On the contrary, if the mean contrast was negative for the most extreme cluster (i.e. assigned as BBB), genotypes were then assigned depending on their mean contrast values in the order BBA, BAA and AAA, from the closest to the furthest cluster from the BBB homozygous genotype (Fig 3).

We assumed that when the outcome of clustering was a single cluster for a given SNP, it could only correspond to a homozygous genotype; 2 or 3 clusters indicated a homozygous genotype and the closest heterozygous or the two heterozygous genotypes; and 4 clusters represented all 4 possible genotypes for triploids. Note that our algorithm can also be used for genotype calling in diploids as the same reasoning could be applied with a maximum of 3 possible genotypes for diploids as long as it is specified in the input parameters to the algorithm.

In case of 3 clusters encountered for a given SNP in triploids, an additional step was added to address the case of a highly shifted signal. This implies markers where genotypes are all shifted toward either positive or negative contrast value making, leading to having a cluster corresponding to a heterozygous genotype in the most extreme position, and thus being wrongly identified as a cluster corresponding to a homozygous genotype. To minimize the error due to that rare behaviour, if the most extreme cluster had less than half the number of individuals as the opposing cluster, it was assigned as a heterozygous genotype, and the opposite cluster was designated as the homozygous genotype (Fig 4, Before to After). In this case however, the next step of the algorithm concerning SNP quality control and decision criterion to retain or remove a SNP would frequently reject the marker. However, we had to first decide

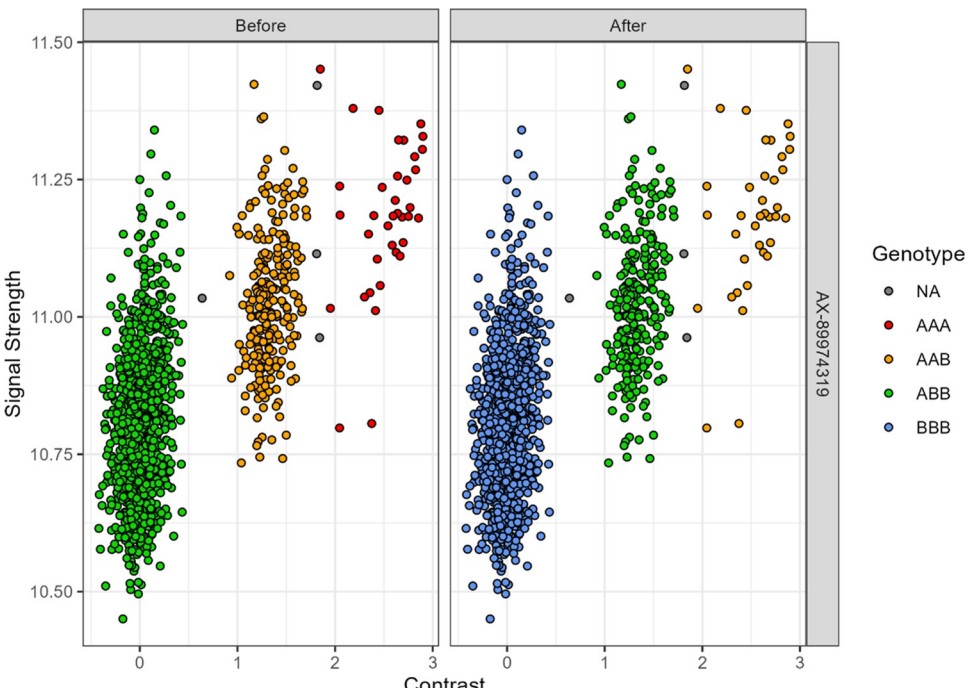

**Fig 4. Example of implementation of the additional step to account for highly shifted contrast signal.**

the most likely genotypes in this case. In a population in which the number of apparent AAA is less than half the number of apparent ABB (equivalent to freqA < 0.55) (ex. Fig 4), the probability to have no BBB in the population (freqB ≥ 0.46) is extremely low as the expected frequency of BBB is ≥ 0.1, i.e. it is more probable that apparent AAA might be an AAB shifted genotype and apparent ABB might be a BBB shifted genotype. In this corrected situation, the frequency of A was less than 0.2 making the AAA genotype extremely rare (with an expected frequency < 0.01 and even not present here) and B higher than 0.8 (explaining the high number of BBB) (Fig 4).

In the situation where samples originate from distinct populations, there is an additional issue to solve for genotype calling when only two clusters are identified for a SNP. In that case, it is likely that the two clusters correspond to the two homozygous genotypes and not to a SNP to be put in the rare category of "No Minor homozygote". Indeed, the SNP is likely to be monomorphic within a given population, but different populations may have fixed alternative alleles.

To solve this case, we used the approach proposed by [26]. We derived reference values for the mean contrasts of all possible genotypes by averaging them across markers with the maximum number of clusters identified (i.e. 4 for triploids). These reference values were used to attribute genotypes for the remaining markers (with a number of clusters below the maximum). For these latter markers, the mean contrast of each cluster was compared to the reference set of values, and the genotype was assigned based on the closest reference value. If two clusters pointed to the same reference value, the genotypes were assigned based on their relative positions. For example, if two clusters pointed toward the negative reference value corresponding to BBB homozygote, the one with the most negative contrast was assigned to the BBB homozygote while the other was assigned to the nearest possible heterozygous genotype BBA.

All the steps of our algorithm (from clustering to genotype calling) can also be used for diploids as the approach can be applied with a maximum of 3 possible genotypes for diploids by indicating the ploidy level of the population under study.

## Quality control for genotypes and SNP categorization

Following the approach proposed by AXAS, seven criteria were employed to enhance cluster precision and identify low-quality markers in the genotype calling phase. Three criteria were used to decide whether or not individuals or clusters were assigned to a given genotype or not assigned (NA):

1) No call for individuals. During clustering, individuals were assigned to a cluster number with a certain probability. Individuals with a probability of belonging to their cluster below 0.85 for a given marker were marked as NA to limit incorrect genotyping. Note that the influence of this threshold on the final result is limited. When merging clusters, we added the probability of the two clusters merged for each individual involved. The probability for an individual to be under 0.85 resulted to be low. We set it to 0.85 as it is the default value in fitPoly, to have comparable results.

2) Distance between individual and its cluster centre. This criterion aimed to avoid wrong genotyping by identifying individuals far from all clusters while still assigned to a cluster. The distance between an individual and the centre of its cluster was monitored to not exceed 2.8 times the standard deviation of the cluster along the Contrast axis ($SD_{cluster}$). An individual genotype was set to NA above this threshold. The choice of a 2.8 factor was based on the property that under the assumption of a normal distribution of individuals within a cluster, 99.5% of the observed values should fall within ±2.8 times the standard deviation. This factor can be modified in the R functions to allow for more flexibility.

3) Cluster Standard Deviation ($SD_{cluster}$). A cluster was set to NA if its $SD_{cluster}$ exceed $0.28*(1+0.5*abs(Mean_{cluster}))$. This criterion imposed a maximal standard deviation to a cluster to limit the risk of genotype calling for a cluster gathering multiple genotypes (in case the algorithm failed to do the correct clustering). The factor of 0.28 was empirically determined through a trial and error assay. The objective was to establish a minimal $SD_{cluster}$ of 0.28 and to progressively increase this minimum as the cluster moved farther away from 0.

The remaining four criteria acted as filters to assess the SNP quality, similar to criteria implemented in the AXAS software, before categorization of the markers:

4) Marker Call Rate (CR). The minimum CR was fixed to 0.97 which is the default value of the AXAS software.

5) Marker Fisher's Linear Discriminant (FLD). The FLD is a measure of the distance between the two nearest genotypes along the x axis (Contrast) and the quality of the clusters. It is defined as:

$$FLD = \frac{abs(Contrast_{Geno1} - Contrast_{Geno2})}{SD_{Geno1,Geno2}} \tag{Eq6}$$

Where $Contrast_{Genoi}$ represented the mean Contrast of genotype i and $SD_{Geno1,Geno2}$ represented the pooled standard deviation of genotype 1 and 2. If the FLD was 3.4 or lower, two genotypes were considered too close to be reliable. This threshold value is an adaptation of the AXAS software one (FLD = 3.6) which was too restrictive especially for polyploids.

6) Marker Heterozygous Strength Offset (HetSO). The HetSO measures the offset between homozygous and heterozygous genotypes along the y axis (Signal Strength). Heterozygous clusters are expected to be positioned higher on the y axis than homozygous clusters (i.e. HetSO value > -0.3).

7) Marker Homozygous Ratio Offset (HomRO). The HomRO represented the position of the homozygous cluster along the x axis (Contrast). The threshold value depended on the number of clusters like so: 0.6, 0.3, 0.3, -0.9 for 1, 2, 3 and 4 clusters, respectively (adapted from [35]).

Markers failing to pass one of these criteria were labelled according to the filter they failed: "Call rate below threshold" for call rate threshold, "Off target variant" for HetSO threshold, and "Others" otherwise. Those are rejected markers, meaning markers with low genotyping confidence that should not be used for further analyses.

Markers passing all four filters were categorized based on their number of genotypes: "Mono high resolution", "No minor homozygote" and "Poly high resolution" for respectively, 1 genotype, 2 or 3 genotypes, and 4 genotypes. Those are accepted markers, meaning markers with high genotyping confidence that could be used for further analyses.

## Comparison strategy between GenoTriplo and fitPoly

To evaluate the efficiency of our method in contrast to an existing alternative, we conducted a comparative analysis between GenoTriplo and fitPoly, the sole package available on the CRAN that handles triploid genotyping.

First, we assessed the overall concordance between GenoTriplo and fitPoly by comparing the genotypes assigned by both methods per individual and marker. Then, we examined the number of genotypes identified by each method for all markers and categorized markers by a pair of integers representing the respective number of genotypes identified by GenoTriplo and fitPoly (for instance category (2;3) corresponded to 2 genotypes found by GenoTriplo and 3 by fitPoly) separating markers in 16 categories.

Categories of equal integer pair (both methods found the same number of genotype) were visually and numerically compared based on the overall genotype concordance rate and the mean contrast value of each genotype for the 4 corresponding categories from (1;1) to (4;4). For the visual comparison, mean cluster position of each genotype for each marker was displayed on a graph to compare genotype global position for each 4 categories.

The genotypes given by GenoTriplo and fitPoly were compared marker-by-marker and the best one was noted based on human visual observation. This was done for all markers in categories gathering 200 or more markers except when both methods found the same number of genotypes. Among the 12 remaining categories, 8 were analysed.

For categories exceeding 1,000 markers, a subset of 1,000 random markers was retained for visual inspection.

For these 8 tested categories, we compared markers acceptance (when a marker passed all quality threshold) and rejection (when a marker did not reach all quality threshold) by the methods to identify any differences. For each category, markers were split into two groups according to the best method to genotype them (GenoTriplo or fitPoly) and an overall genotype concordance rate between the two methods for all the 16 categories was computed.

Both methods had high marker call rate on average (0.98 (± 0.044) for GenoTriplo and 0.97 (± 0.122) for fitPoly). To ensure fair comparison, all NA were removed and not considered as different between methods, recognizing that some NA may be attributed for quality purpose when samples did not clearly belong to a genotype while others may result from misidentification of clusters by one or the other method. This approach aimed to provide a robust comparison while considering the nuances of missing data especially for those methods that provided few NA.

## Parentage assignment assessment

To validate the utility of GenoTriplo, we conducted a parentage assignment of the triploid individuals using the R package APIS with the newly available function that enables parentage assignment on triploids [36]. The assignment was done using the 1,000 best markers selected based on their Minor Allele Frequency (MAF) and CR. These markers were chosen from the 32,325 markers that successfully passed through all applied filters, including "Poly high resolution", "Mono high resolution" and "No minor homozygote".

While the true parents of the offspring were not available to fully validate the parentage assignment, we had access to the mating plan, which is composed of 10 independent factorial matings, each being composed of 8 to 10 sires crossed one-by-one with 17 to 24 dams, producing a theoretical number of 1862 full-sib families (or 1862 valid parent pairs). However, parental assignment by exclusion considers all possible parental pairs from the 98 sires and 190 dams [36, 37], and thus a theoretical number of 98*190 = 18620 possible parent pairs, which is 10 times more than the valid ones. In case of inaccurate assignments, we would thus expect that approximately 9 out of 10 would fall out of the declared mating plan.

## Validation dataset

Thanks to the authors of [38], we were able to test our method on a different dataset. From a file given by the authors upon request, signals of A allele and B allele presence for 18012 SNP markers in an Illumina SNP array were extracted for 68 triploid apple cultivars. Even though the genotyping technology was different, those signals were resembling the luminescence data required by GenoTriplo. The genotypes of those 68 individuals for 10295 markers were available at https://www.rosaceae.org/publication_datasets, reference number tfGDR1061 [38]. They were obtained with ploidyClassifier, a custom Python script [38]. We compared the genotypes found by GenoTriplo with the genotype obtained by [38] using ploidyClassifier.

## R package and shiny application

For enhanced accessibility, we developed a R package called 'GenoTriplo' available on CRAN. The package incorporates functions for executing both the clustering phase ('Run_Clustering') and the genotype calling phase ('Run_Genotyping'). Additionally, to make the usage easier for beginners and experts, a shiny interface was implemented ('launch_GenoShiny'), organized into four steps.

First, the raw dataset from AXAS requires formatting before progressing through the clustering phase. A list of markers or/and a list of individuals can be provided to select specific markers or/and individuals.

The clustering phase starts with the refined dataset obtained at the previous step. Users are prompted to input the ploidy level (default set to 3) of the population and the number of cores for parallelization (default set to $N_{computer\_cores}$-2). An option to fine-tune parameters is available through the 'Add more control' button, allowing adjustments of the number of initializations for the Rmixmod clustering function (default set to 5) and the minimal contrast distance between two clusters (default set to 0.28).

The genotype calling process is applied to the output of the clustering phase. Users have the option to provide a CSV file containing the correspondence between A/B signals of AXAS and ATCG bases. Inputs such as the ploidy of individuals (default set to 3), the number of cores for parallelization (default set to $N_{computer\_cores}$-2), and whether or not individuals originate from the same population are requested (default set to same population). The latter is introduced for simplification, assuming that individuals from the same population cannot exhibit both homozygous genotypes without a heterozygote (as described in **Genotype Calling** section).

This step provides flexibility with various adjustable parameters, including no-call threshold for individuals, distance between cluster centres, cluster standard deviation threshold, FLD threshold, HetSO threshold, and CR threshold for markers.

The final step is optional and enables users to visualize the genotyping results through graphs and statistics. During this optional step, users can select individuals on the graphs to manually change their genotypes and save the changes.

All graphics were made using ggplot2 [39] via R code [34].

## Results

### Clustering and genotype calling phases

A "Poly high resolution" marker was characterized by the maximum number of genotype and well-separated clusters (Fig 5, "Poly high resolution") whereas a "No minor homozygote" marker shared these characteristics but lacked one of the homozygous genotypes (Fig 5, "No minor homozygote"). Occasionally, despite apparent separation of clusters, they failed to meet all established thresholds and did not pass filters; for example, in Fig 5, "Others (FLD threshold)" exhibited a FLD of 3.36, slightly below the set threshold of 3.4.

The methodology demonstrated robustness in identifying issues related to the position of the heterozygote cluster (Fig 5, "Off target variant"), where the BBA genotype exhibited lower signal strength than the BBB genotype, and in detecting mixed or uncertain clusters by augmenting the number of NA among individuals between clusters (Fig 5, "Call rate below threshold").

### Number of initializations and maximal number of clusters

To assess the impact of the numbers of initializations and of maximum clusters in GenoTriplo, we conducted a quantitative comparison of the marker distribution across various categories following the completion of the clustering and genotyping phases.

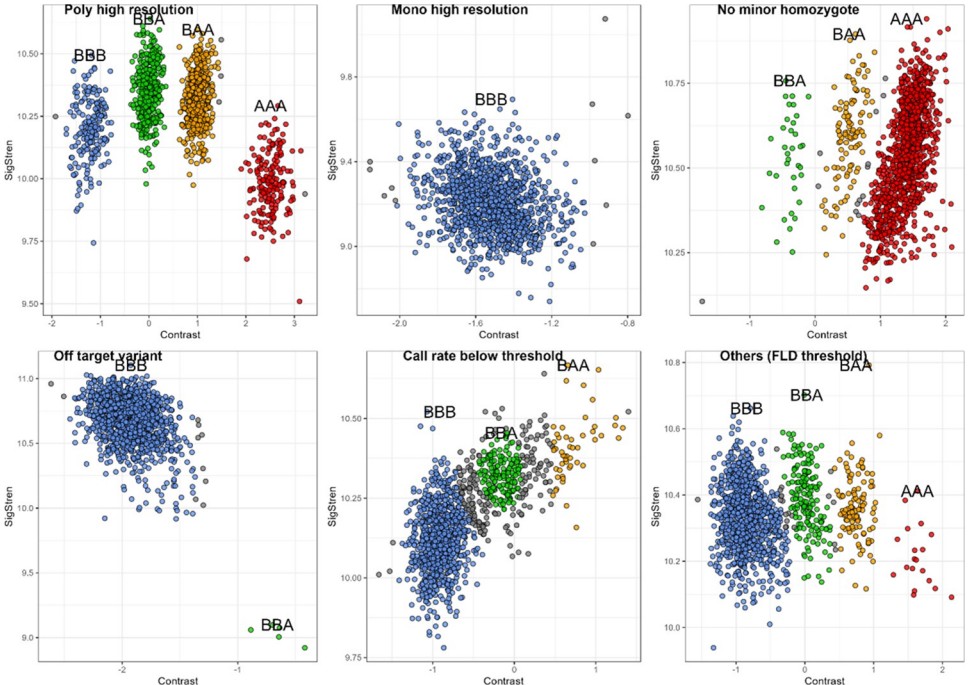

**Fig 5. Examples of distribution on the axes of contrast and signal strength of genotypes identified by GenoTriplo for each category of markers.**

**Table 1. Number of markers by categories for the different parameters used in clustering phase.**

| Runs | | Categories | | | | | |
|---|---|---|---|---|---|---|---|
| $N_{init}$ | $N_{clus}$ | Poly high resolution | No minor homozygote | Mono high resolution | Call rate below threshold | Off target variant | Others |
| 1 | 8 | 18307 | 7126 | 4315 | 7451 | 411 | 423 |
| 5 | 8 | 21715 | 6233 | 4377 | 4734 | 421 | 553 |
| 10 | 8 | 22501 | 5838 | 4299 | 4344 | 438 | 613 |
| 5 | 4 | 11867 | 8480 | 4612 | 12513 | 400 | 161 |
| 5 | 12 | 22875 | 4452 | 3132 | 6516 | 403 | 655 |

The number of initializations positively impacted the performance of the algorithm. The number of markers in "Poly high resolution" category increased steadily from 1 to 10 initializations (+18% from 1 to 5 and +5% from 5 to 10), while numbers in "No minor homozygote" and "Call rate below threshold" categories decreased. The supplementary "Poly high resolution" markers identified with 10 initializations, compared to 5, originated partly from the "Call rate below threshold" category. This subset of markers may have encountered call rate issues due to cluster standard deviation thresholds. If the low-frequency genotype was not found, it might have been erroneously grouped with another genotype, significantly increasing the standard deviation of the cluster and resulting in NA assignments for all individuals in that cluster. Another subset originated from the "No minor homozygote" category, where individuals belonging to a smaller, low-frequency genotype might have been inaccurately grouped with a higher frequency genotype. This led to a lesser increase in standard deviation or NA assignments due to the distance-to-centre threshold. "Others" category showed less sensitivity to changes in the number of initializations (Table 1).

Increasing the initial number of clusters defined for Rmixmod clustering function also helped to get more markers included in the "Poly high resolution" category, especially when increasing from 4 to 8 clusters and, to a lesser extent, from 8 to 12 clusters (Table 1). Conversely, the number of SNPs in the "No minor homozygote" category decreased, respectively from 8,480 to 4,452 markers with 4 and 12 initial clusters, respectively. Notably, the number of markers in the "Mono high resolution" category decreased substantially for 12 clusters (3,132), while it remained stable around 4,300 for 4 and 8 initial clusters. The number of markers in the "Call rate below threshold" category strongly decreased from 4 to 8 initial clusters (12,513 to 4,734), but increased from 8 to 12 initial clusters (4,734 to 6,516), indicating an optimal number of initial clusters of 8 as compared to 4 and 12 clusters. Although the number of SNPs put in "Others" category increased with the number of clusters, it did not counterbalance the decrease of SNPs in "Call rate below threshold" category, indicating that some markers were pulled out of the low-quality categories towards the high-quality categories (Table 1).

In summary, utilizing 5 initializations, 8 clusters, and default parameters and thresholds for quality control of the genotyping resulted in 85% of markers falling into high quality marker categories i.e. "Mono high resolution", "No minor homozygote" and "Poly high resolution".

## Comparison between GenoTriplo and fitPoly genotyping

The overall concordance rate between genotypes derived from GenoTriplo and fitPoly was 85%, reaching 89% after exclusion of all NA. Notably, 26% of the SNPs showed differences in the number of genotypes identified by the two methods. GenoTriplo found less SNPs with four genotypes, while fitPoly found less monomorphic SNPs (Table 2).

In categories for which both GenoTriplo and fitPoly identified the same number of genotypes, the genotype concordance was not as high as expected. For a single genotype found, the

**Table 2. Table with the respective number of SNPs with 1, 2 3 or 4 genotypes identified with GenoTriplo or fitPoly.**

| GenoTriplo\fitPoly | 1 genotype | 2 genotypes | 3 genotypes | 4 genotypes |
|---|---|---|---|---|
| 1 genotype | 2333 | 1429 | 644 | 86 |
| 2 genotypes | 28 | 493 | 542 | 783 |
| 3 genotypes | 28 | 210 | 2289 | 4333 |
| 4 genotypes | 38 | 640 | 966 | 23001 |

concordance was 25%, increasing to 81% with two genotypes found, 94% with three genotypes found, and exceeding 99% with four genotypes found. The difference in the case of a unique genotype assigned was due to fitPoly frequently assigning a heterozygous genotype rather than a more likely homozygous genotype. Out of 2428 markers with a single genotype assigned by fitPoly, 1752 were identified as heterozygous (Fig 6).

A similar pattern emerged, to a lesser extent, when fitPoly identified two genotypes. In contrast, GenoTriplo exhibited the expected behaviour, with each distinct genotype forming distinct clusters, displaying distinct mean contrast values regardless of the number of genotypes identified (Fig 6).

When the numbers of possible genotypes were different across the two methods, two discernible patterns emerged from the analysis based on visual observation of the clusters, showcasing scenarios where fitPoly outperformed GenoTriplo and *vice versa* (Table 3). FitPoly showed better results in categories where it identified a greater number of genotypes compared to GenoTriplo, specifically in categories (2;3), (2;4), and (3;4). For these 3 categories however, the genotypes provided by GenoTriplo closely matched those from fitPoly when ignoring NA calls, with concordance rates of 99%, 99%, and 97%, respectively. Notably, for 292 markers out of the 1,000 assessed in the (3;4) category, fitPoly identified a lone individual for the minor homozygous genotype, which GenoTriplo categorized as NA.

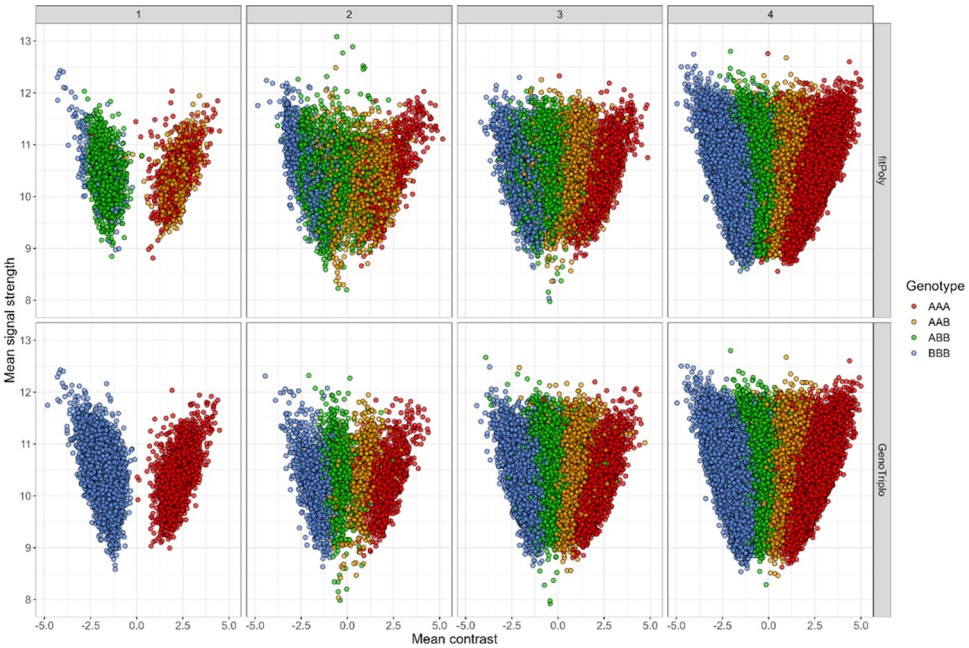

**Fig 6. Mean contrast and signal strength values for genotypes of SNP with 1, 2, 3 and 4 different genotypes (from left to right) for fitPoly (above) and GenoTriplo (under) methods.**

**Table 3. Number of markers visualized per category, number best genotyped by GenoTriplo, by fitPoly; and corresponding rate of concordant genotypes between methods.**

| Category (GT; FP) | Total visual observation | Number of markers | | | Rate of concordant genotypes for markers with | |
|---|---|---|---|---|---|---|
| | | Best genotyping: GenoTriplo | Best genotyping: fitPoly | No best method or bad marker | Best genotyping: GenoTriplo | Best genotyping: fitPoly |
| (1;2) | 1000 | 946 | 6 | 48 | 0.49 | 1 |
| (1;3) | 644 | 330 | 282 | 32 | 0.30 | 1 |
| (2;3) | 542 | 61 | 354 | 127 | 0.69 | 0.99 |
| (2;4) | 783 | 37 | 657 | 89 | 0.60 | 0.99 |
| (3;2) | 210 | 126 | 5 | 79 | 0.49 | 0.78 |
| (3;4) | 1000 | 105 | 784 | 111 | 0.89 | 0.97 |
| (4;2) | 640 | 582 | 0 | 58 | 0.34 | - |
| (4;3) | 966 | 841 | 50 | 75 | 0.40 | 0.72 |

Conversely, in categories where GenoTriplo exhibited superior performance (categories (1;2), (3;2), (4,2), and (4,3)), fitPoly's genotypes deviated significantly from the expected outcomes, resulting in concordance rates of 49%, 49%, 34%, and 40%, respectively.

In the (1;3) category, a balanced performance between the two methods was observed. When fitPoly outperformed, GenoTriplo's genotypes closely matched fitPoly's (achieving 100% concordance after removing all instances of "NA"). However, when GenoTriplo was better, only 30% of fitPoly's genotypes aligned with the decisions made by GenoTriplo.

When examining the SNP acceptance/rejection categorization, we found that GenoTriplo retained the majority of SNPs where fitPoly performed better, aligning with expectations due to the close similarity between GenoTriplo and fitPoly. However, most SNPs within the (2;4) category were rejected by GenoTriplo and no by fitPoly, particularly for call rate considerations. Notably, most markers rejected by fitPoly in these categories were also rejected by GenoTriplo.

In the case of SNPs where GenoTriplo exhibited superior performance, fitPoly retained nearly half of them, despite having low concordance with GenoTriplo. For instance, in the (1;2) category, out of the 936 SNPs retained by GenoTriplo, 632 were also retained by fitPoly, even though they were likely incorrect, given the 50% concordance with GenoTriplo. Notably, almost every SNP rejected by GenoTriplo was also rejected by fitPoly.

## Parentage assignment assessment by APIS

To evaluate the genotyping performance for pedigree retrieval, we utilized the exclusion method of APIS (https://cran.r-project.org/web/packages/APIS/index.html) for parentage assignment of triploid offspring genotyped with the described method [36], alongside parents genotyped by AXAS software. All offspring were successfully assigned to a couple of parents belonging to the correct factorial mating plan.

For the best couples assigned, a maximum of 19 mismatches occurred among the 1000 markers, with a mean mismatch of 6.9, representing less than 1% of mismatches between parents and progeny (Fig 7). The second-best couples exhibited a minimum of 47 mismatches, with a mean of 85.6. Therefore, a substantial gap in mismatch numbers existed between the best and second-best couples, with distributions clearly exhibiting no overlap, showing the very high quality of the assignments obtained (Fig 7).

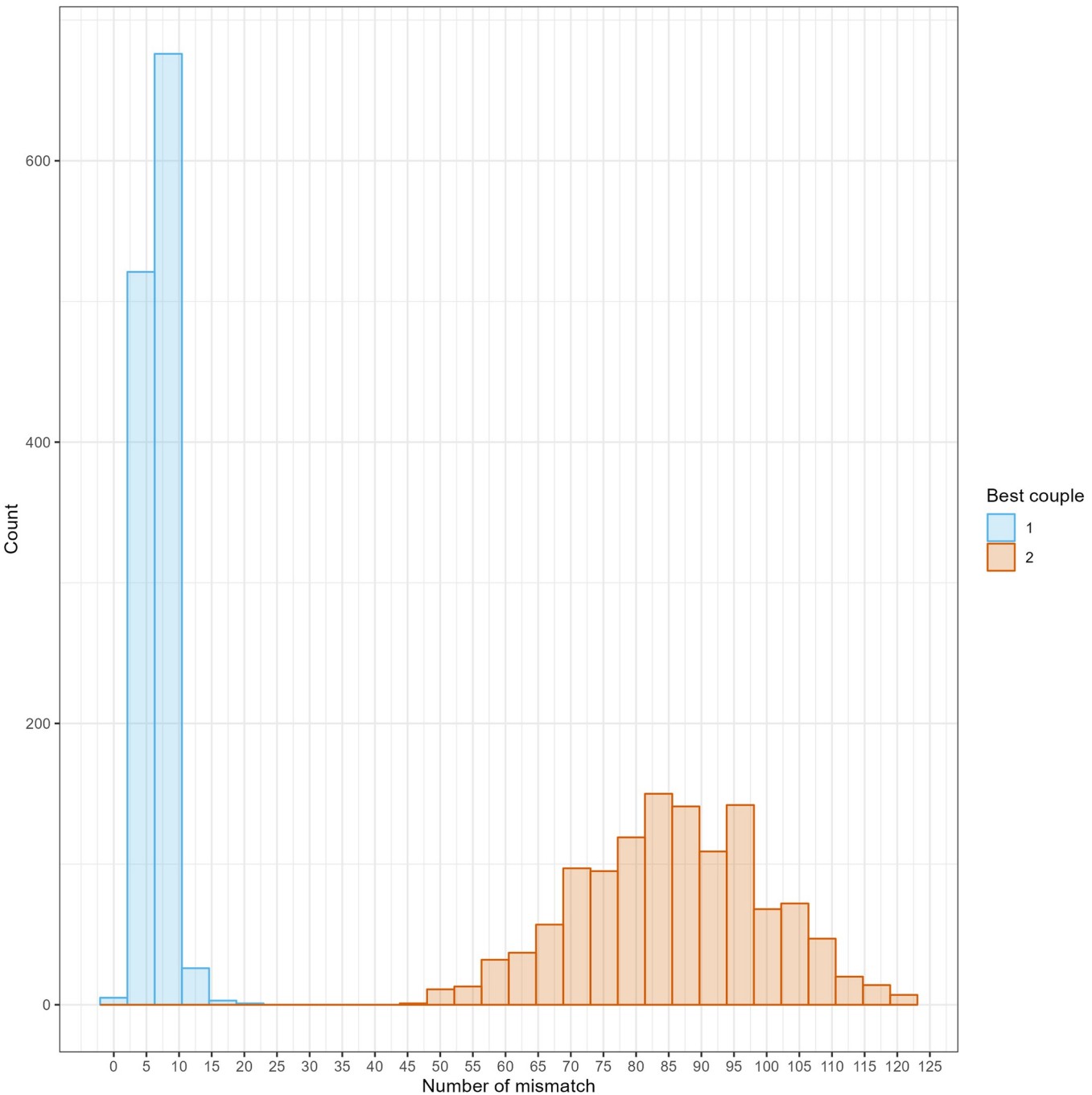

**Fig 7. Number of offspring as a function of the number of mismatches for the best couple (blue) and the second-best couple (red) found by APIS parentage assignment.**

## Validation results on an apple triploid dataset

The concordance rate between GenoTriplo genotypes and the published genotypes for apple triploids (obtained by ploidyClassifier) was low (69.96%) for the 10295 markers. This was mainly due to NoCall genotypes. By removing the NoCall genotypes, the concordance rate increased to 98.76% between the two genotyping methods. The overall call rate by GenoTriplo

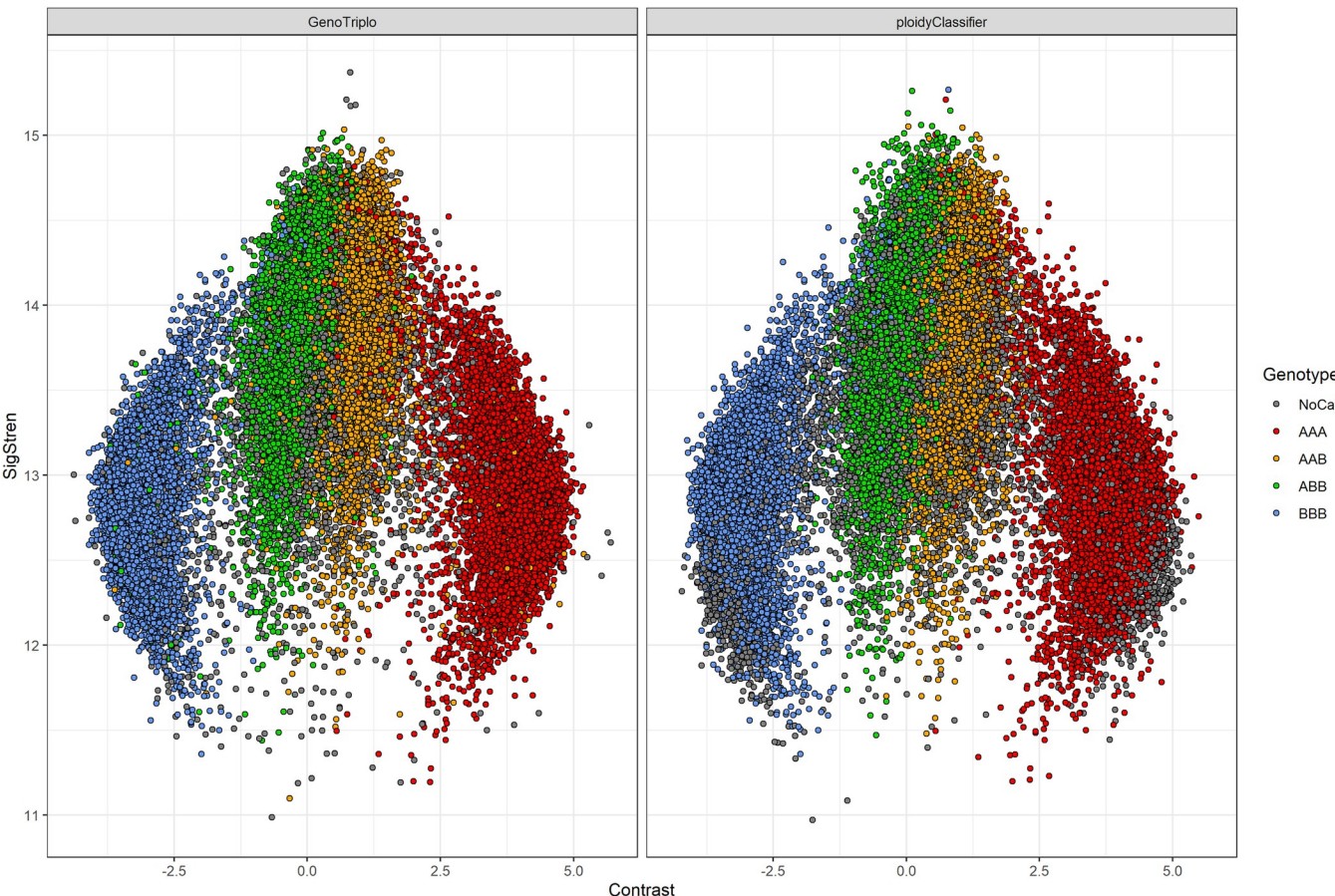

**Fig 8. Mean Contrast and SigStren value for each marker genotype given by either GenoTriplo or ploidyClassifier.**

was 94.82% against 88.97% for the published genotypes. GenoTriplo performed well (Fig 8) with well discriminated genotypes even though the raw luminescence data type was not the one it was developed with (Thermofisher for GenoTriplo, Illumina for ploidyClassifier). The main difference was due to the higher number of NoCalls with ploidyClassifier.

We tested the potential impact of Dmin value on the apple genotyping results with Geno-Triplo using the available 18k markers. We found an average concordance rate of 99.5% for Dmin values ranging from 0.23 to 0.32 when comparing with the run using our default value of 0.28 by keeping only high-quality markers from the compared runs. More markers were retained as the Dmin used increased from 0.23 to 0.28, peaking at around 10k markers out of the 18k. This number remained stable up to a Dmin value of 0.32. Additionally, we recalculated Dmin as explained in the clustering algorithm section of the Materials and Methods by extracting values of mean Contrast and distance from the separation between clusters that should and should not be merged. We also found an average Dmin value equal to 0.28 as it was the case for our trout dataset.

## Computational cost

Memory is directly dependent on the size of the input dataset of luminescence (N Go). It reaches a maximum of 2N Go approximately when loading the initial dataset and computing the different variables. For clustering and genotyping, the dataset is divided to reduce memory

consumption but should not exceed 2N Go. Time depends on multiple parameters: Ncore, Nmarkers, Nindividuals, Ninit especially. For example:

10 cores, 38.000 SNP, 1230 individuals, 1 init: 30min for clustering, 35min for genotyping

10 cores, 38.000 SNP, 1230 individuals, 5 init: 1h45min for clustering, 35min for genotyping

10 cores, 18.000 SNP, 68 individuals, 5init: 3min30 for clustering, 5min30 for genotyping

To reduce memory storage, we kept only three digits for Contrast and SigStren.

## Discussion

Our method for genotype calling of triploids from luminescence datasets demonstrated its quality to genotype triploid fish, leading to its integration into the R package GenoTriplo, freely accessible to the scientific community: https://cran.r-project.org/web/packages/GenoTriplo/index.html.

Our approach demonstrated a good accuracy for parentage assignment of triploid offspring with diploid parents. This was validated using the top 1000 markers based on MAF and Call Rate. The method performed well even with fewer markers or randomly selected markers (as few as 200). Although the true pedigree was unknown, the very low numbers of mismatches for the best couple suggested highly accurate assignments.

The method did not depend on prior information on genotype position relative to their own contrast value when identifying genotypes among SNP. This characteristic enhanced efficiency, particularly when contrast values were shifted from the expected values as a same genotype would manifest at different value of contrast dependant on the marker [24,25]. This also allows to genotype new SNPs with no need of human action to set reference genotypes for each SNP, in this way differentiating it from AXAS that relies on reference genotype.

The clustering method underlying the genotyping call was efficient using well-fitted input parameters. Notably, the number of initializations significantly enhanced the clustering algorithm's efficiency by identifying clusters with few individuals, i.e. representing low-frequency genotypes. In our case study, the occurrence of markers with low-frequency genotypes was limited, and most of the different genotypes were thus well-identified with only 5 initialization runs.

Increasing the number of initializations will maximize the probability of identifying clusters corresponding to low-frequency genotypes. However, this increase results in longer computation time, forcing a trade-off between computation time and additional identification of very low-frequency genotype for few SNPs. In our case, using 5 initializations was a good compromise, but this parameter should be optimized for other triploid populations and species.

In addition, the initial number of clusters also significantly influenced the clustering algorithm outcomes. Requesting only 4 clusters for triploids resulted in miss-detection of low-frequency genotypes, leading to a shortage of "Poly high resolution" SNPs and an excess of "No minor homozygote" markers. Conversely, too high a number of clusters led to inappropriate creation of clusters composed of very few individuals, and resulting in a scarcity of the "Mono high resolution" category. Optimal results were achieved with an intermediate number of clusters, specifically twice the number of possible genotypes (8 for triploids). This configuration allowed for the identification of most of the low-frequency genotypes without generating artefacts. Therefore, our strategy using twice the maximum number of possible genotypes facilitated genotype calling for low-frequency genotypes without the need for of large number of individuals to genotype together as suggested by [23,24].

In the genotyping process, the method employed assumed that individuals originated from the same population. Using Hardy-Weinberg hypothesis, our approach did not accept that both homozygous genotypes coexisted without the two heterozygous genotypes for a given SNP, contributing to the efficiency of our genotype attribution. When informed that the samples can come from various populations, our method involved the comparison of mean contrast values of each current cluster to the values of reference clusters. Those reference values are derived on the same dataset from markers with the maximum number of genotypes. Given the common occurrence of contrast value shifts (when all contrast values of a SNP are all shifted toward positive or negative value), the recommended approach, when possible, is to analyse together pools of individuals originated from the same population.

The overall concordance of genotypes between GenoTriplo and fitPoly was notably high. However, differences emerged when comparing the number of genotypes identified by each method. When both methods identified the same number of genotypes, differences were the result of the fundamentally different approaches to assigning genotypes to clusters of individuals. GenoTriplo relied on stringent assumptions, like assigning a homozygous genotype when only one cluster was identified. In contrast, fitPoly lacked such guidelines, leading to substantial discordance, especially in cases where only one genotype was expected.

GenoTriplo encountered difficulties in identifying all 4 genotypes, often settling for 3 when very few individuals formed the second homozygous genotype. Those few individuals usually were not assigned a genotype, avoiding genotyping errors. Besides, for 292 markers among the 784 markers where fitPoly identified 4 genotypes while GenoTriplo found only 3, a single individual represented the homozygous low frequency genotype in FitPoly. The credibility being low for a single individual to represent a genotype, we consider it preferable to assign the individual to NA, thus avoiding a possible genotyping error.

On the contrary, fitPoly faced difficulties in identifying a limited number of genotypes (below the maximum possible) for a given SNP, particularly when the SNP was monomorphic. This challenge could come from the method per se which prioritizes a high number of genotypes, leading to the creation of unwanted clusters. While some of these SNP were rejected by fitPoly for excess of NA, half were retained even for those with low concordance with GenoTriplo, causing substantial genotyping errors.

While most of the disagreement were minor when fitPoly performed better, GenoTriplo's accuracy outperformed fitPoly's, especially for low number of genotypes and detection of wrong genotypes.

This paper focuses on the genotyping of triploids, but it is essential to note that the method was also successfully tested on diploids, providing similar results to the AXAS software. Furthermore, its application could potentially be extended to higher ploidy levels. The key parameter for the clustering phase would be the minimal distance between two clusters. Notably, the mean contrast value for a homozygous diploid genotype matched that of a triploid homozygous genotype. Consequently, with higher ploidy levels, the insertion of additional heterozygote genotypes is expected between the contrast values of homozygotes, resulting in diminishing distances between clusters as ploidy levels increase, making the discrimination between different allelic dosages more difficult. Currently, the genotyping phase is implemented for diploid and triploid individuals, and further work would be required to extend it to higher ploidy levels.

GenoTriplo was extensively tested on luminescence data from AXAS software but also performed well with Illumina luminescence data, even though in this case only 68 validation individuals were available, albeit with 18000 markers each.

## Acknowledgments

Authors are indebted to Dominique Charles, Alexandre Desgranges and Jean Ruche from the Aquaculteurs Bretons trout selective breeding company (Plouigneau, France) for the fish production, and to Charles Poncet and his team for the genotyping of all fish for HypoTemp project. We also sincerely thank Hélène Muranty for providing the apple validation dataset, and the following groups and persons for providing the corresponding apple accessions: Matthew Ordidge, who provided samples from the UK National Fruit Collection, members of the following French amateur associations and conservatories who maintained the plant material and provided samples: Les Croqueurs de pommes, Les Mordus de la pomme, Centre Végétal Régional d'Aquitaine (CVRA), Centre Régional de Ressources Génétiques des Hauts-de-France (CRRG), Société Pomologique du Berry, I z'on creuqué eun' pomm', Fédération Départementale Variétés Locales 12, Jardin du Luxembourg, the INRAE Biological Resource Center "Pome Fruits and Roses" (RosePom: https://www6.angers-nantes.inrae.fr/irhs/Ressources-mutualisees/Ressources-genetiques/CRBFruits-a-pepins-et-rosier), and the INRAE Horticulture Experimental Facility (UE HORTI: https://doi.org/10.15454/1.5573931618268674E12).

## Author Contributions

**Conceptualization:** Mathieu Besson, François Allal, Pierrick Haffray, Pierre Patrice, Marc Vandeputte, Florence Phocas.

**Data curation:** Julien Roche.

**Formal analysis:** Julien Roche.

**Funding acquisition:** Pierrick Haffray, Florence Phocas.

**Investigation:** Julien Roche.

**Methodology:** Julien Roche, Mathieu Besson, François Allal, Pierre Patrice, Marc Vandeputte, Florence Phocas.

**Project administration:** Pierre Patrice, Florence Phocas.

**Resources:** Pierre Patrice.

**Software:** Julien Roche.

**Supervision:** Mathieu Besson, Florence Phocas.

**Validation:** Julien Roche.

**Visualization:** Julien Roche.

**Writing – original draft:** Julien Roche, Florence Phocas.

**Writing – review & editing:** Mathieu Besson, François Allal, Pierrick Haffray, Pierre Patrice, Marc Vandeputte.

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
