## [Decision Letter · Decision Letter 0]

1 Jul 2024

Dear Dr Phocas,

Thank you very much for submitting your manuscript "GenoTriplo: A SNP genotype calling method for triploids" for consideration at PLOS Computational Biology.

As with all papers reviewed by the journal, your manuscript was reviewed by members of the editorial board and by several independent reviewers. In light of the reviews (below this email), we would like to invite the resubmission of a significantly-revised version that takes into account the reviewers' comments.

Two reviewers raised significant concerns about the current manuscript. The authors are highly suggest to revise the manuscript to resubmit a revised manuscript or resubmit it as new submission with the original manuscript ID.

We cannot make any decision about publication until we have seen the revised manuscript and your response to the reviewers' comments. Your revised manuscript is also likely to be sent to reviewers for further evaluation.

Sincerely,

Feixiong Cheng, Ph.D.

Academic Editor

PLOS Computational Biology

Ilya Ioshikhes

Section Editor

PLOS Computational Biology

Two reviewers raised significant concerns about the current manuscript. The authors are highly suggest to revise the manuscript to resubmit a revised manuscript or resubmit it as new submission with the original manuscript ID.

Reviewer's Responses to Questions

**Comments to the Authors:**

Reviewer #1: In this paper, the authors developed GenoTriplo, a computational method to call SNP genotype for triploids. This work addressed an important research question. The proposed methods are interesting. The authors conducted comprehensive benchmark analysis to compare GenoTriplo with existing method fitPoly, and results are solid. The paper is also well written. I think this work has potential to attract general interest of the readers of PLOS Computational Biology. I just have a few minor comments and suggestions which may further improve the current manuscript. Below are my specific comments:

1. The authors compared the performance between GenoTriplo and fitPoly. When two methods provide inconsistent genotype calling results, what is the gold standard / grand truth? In practice, when the genotype truth is unknown, how can users evaluate the quality of GenoTriplo called genotypes? It would be very helpful if the authors can provide some general guidance for practitioners.

2. In Page 6, line 143. The coefficient 0.28 likes very ad hoc. The authors need to perform robustness analysis, and provide more details on the "trial and error assay" (line 151). When applying to different datasets, whether users need to change this coefficient 0.28? Similarly, the authors need to provide more justification on the following numbers: (1) Page 11, line 228 "0.85", (2) Page 12, line 245 "0.97", (3) Page 12, line 250 "3.4"

3. The authors developed R package GenoTriplo and released it at CRAN. They also need to provide more details on the computational cost, including memory and time.

4. Typo: Page 2, line 36. "One can utilize individuals" is duplicated.

Reviewer #2: The SNP genotype calling method (GenoTriplo) developed for triploids is very interesting. However, the following concerns should be addressed.

Comments:

1.To better test the accuracy of genotyping calling using GenoTriplo, the authors should test the genotype calling from various populations of genetically distant and compare the results with that of single population analysis performed in the current study.

2.Table 2 shows that GenoTriplo identifies significantly less SNPs compared to fitPoly with increased complexity of genotypes. It indicates that GenoTriplo is less efficient in case of complex genotypes. Please provide possible explanation if possible with analysis, for using GenoTriplo over fitPloly.

3.Please correct repetitive sentences from the author summary section.

**Have the authors made all data and (if applicable) computational code underlying the findings in their manuscript fully available?**

Reviewer #1: Yes

Reviewer #2: None

PLOS authors have the option to publish the peer review history of their article (what does this mean?). If published, this will include your full peer review and any attached files.

Reviewer #1: No

Reviewer #2: No
---

## [Decision Letter · Decision Letter 1]

17 Aug 2024

Dear Dr. Phocas,

Thank you very much for submitting your manuscript "GenoTriplo: A SNP genotype calling method for triploids" for consideration at PLOS Computational Biology. As with all papers reviewed by the journal, your manuscript was reviewed by members of the editorial board and by several independent reviewers. The reviewers appreciated the attention to an important topic. Based on the reviews, we are very happy to further consider this manuscript for publication, providing that you fully modify the manuscript according to the review recommendations.

Although the authors addressed major critiques from both reviewers, additional concerns should be fully address before the final editorial decision.

Sincerely,

Feixiong Cheng, Ph.D.

Academic Editor

PLOS Computational Biology

Ilya Ioshikhes

Section Editor

PLOS Computational Biology

Although the authors addressed major critiques from both reviewers, additional concerns should be fully address before the final editorial decision.

Reviewer's Responses to Questions

**Comments to the Authors:**

Reviewer #1: The authors did a good job in the revision, and have fully addressed my previous comments and suggestions. The revised manuscript has been significantly improved. I don't have additional comments and recommend the current version for acceptance.

Reviewer #2: I appreciate the authors for improving the manuscript.

**Have the authors made all data and (if applicable) computational code underlying the findings in their manuscript fully available?**

Reviewer #1: Yes

Reviewer #2: None

PLOS authors have the option to publish the peer review history of their article (what does this mean?). If published, this will include your full peer review and any attached files.

Reviewer #1: No

Reviewer #2: No

Figure Files:

Data Requirements:

Reproducibility:

References:

---

## [Editor Report · Decision Letter 2]

12 Sep 2024

Dear Dr. Phocas,

We are pleased to inform you that your manuscript 'GenoTriplo: A SNP genotype calling method for triploids' has been provisionally accepted for publication in PLOS Computational Biology.

Best regards,

Feixiong Cheng, Ph.D.

Academic Editor

PLOS Computational Biology

Ilya Ioshikhes

Section Editor

PLOS Computational Biology

---

## [Editor Report · Acceptance letter]

17 Sep 2024

PCOMPBIOL-D-24-00412R2 

GenoTriplo: A SNP genotype calling method for triploids

Dear Dr Phocas,

I am pleased to inform you that your manuscript has been formally accepted for publication in PLOS Computational Biology. Your manuscript is now with our production department and you will be notified of the publication date in due course.

With kind regards,

Anita Estes
